# Ego-VGA: A Compact Multimodal Assistant for Egocentric Video–Grounded Reasoning

## Abstract

Egocentric AI assistants have emerged as a promising paradigm for real-world human–AI interaction, yet existing approaches face a critical trade-off: large language models provide strong reasoning but are too resource-intensive for mobile deployment, while lightweight models remain text-centric and lack interactive visual grounding. We introduce Ego-VGA, a lightweight multimodal assistant that delivers goal-oriented visual guidance with high efficiency. Ego-VGA incorporates a novel multimodal fusion layer, where region fusion supports fine-grained vision–language grounding and vision fusion distills temporal cues from egocentric video streams for context-aware reasoning. A lightweight projection module and a compact LLM further enhance efficiency, enabling deployment on mobile and wearable devices. To foster research in intent modeling, we construct Ego-IntentBench, a challenging benchmark with fine-grained procedural annotations. Extensive experiments validate our approach: Ego-VGA achieves +8.7% recall@1 on AssistQ, +17.2 BLEU-1 / +7.6 METEOR on YouCook2, and 20% mean top-5 recall improvement on MECCANO (RGB-only). On Ego-IntentBench, where strong baselines such as Qwen2.5-VL and MiniCPM-V4 degrade substantially, Ego-VGA consistently outperforms them, demonstrating state-of-the-art generalization and adaptability in complex, goal-directed reasoning under visual guidance. The code and dataset are available at `https://anonymous.4open.science/r/Ego-VGA-05CC`

## 1 Introduction

Egocentric AI assistants (Plizzari et al., 2024; Li et al., 2025; Yang et al., 2025) are rapidly emerging as a promising direction for real-world human–AI interaction. Unlike traditional assistants (Graves et al., 2013; Hoy, 2018), such as Siri or Alexa, which primarily rely on text or speech interfaces, egocentric assistants leverage first-person perception and multimodal integration to infer user intent and provide context-aware and real-time guidance in complex environments. This capability is particularly valuable when users face unfamiliar tasks (Damen et al., 2018; Wong et al., 2022), such as cooking a new recipe or operating an unfamiliar device. In such scenarios, users typically consult instructional videos. Egocentric assistants, however, can go further—recognizing objects and environment states, extracting key procedural steps from demonstrations, and overlaying visual cues (e.g., bounding boxes) directly onto the user's field of view. This creates a "what you see is what you get" paradigm, enabling immersive skill transfer and actionable task guidance.

Recent progress in AI assistants largely follows two distinct trajectories. The first focuses on large language models (LLMs), which excel at complex reasoning, multi-turn dialogue, and cross-task generalization. GPT (Radford et al., 2018; Ouyang et al., 2022) demonstrates strong generative capabilities, LLaMA (Touvron et al., 2023a;b) provides efficient scaling and few-shot generalization, and Gemini (Team et al., 2024; Comanici et al., 2025) integrates multimodal contextual understanding. While powerful, these models demand heavy computation and memory, making them impractical for mobile or wearable deployment.

The second trajectory focuses on lightweight models optimized for resource-constrained environments. Examples include MobileLLaMA (Wong et al., 2022), which achieves efficient inference via compression and quantization; LLaVA-mini (Zhang et al., 2025), which employs token reduction and lightweight projectors for multimodal acceleration; and Vinci (Huang et al., 2024b), which

uses efficient multimodal adapters. These approaches are easier to deploy but remain constrained: they are often text-centric, lack strong visual grounding, and cannot provide interactive visual guidance. Taken together, these two directions reveal a critical gap: there is still no practical egocentric assistant that unifies vision–language grounding, real-time interactive guidance, and lightweight efficiency suitable for mobile deployment.

**Contribution.** To address this gap, we propose **Ego-VGA**, a lightweight multimodal egocentric assistant purpose-built for goal-oriented visual guidance. Ego-VGA introduces a novel multimodal fusion framework. Specifically, region fusion integrates object features with their spatial positions and embeds them into the text space, enabling object-level grounding for precise interactive guidance. In parallel, vision fusion distills temporal context from noisy egocentric video streams while compressing the visual input representation, thereby supporting more efficient context-aware reasoning. To further enhance efficiency, we design a lightweight and stable projection layer for multimodal alignment, employ a unified vision encoder to process both images and videos, and leverage a compact yet high-performance LLM as the reasoning engine. These design choices collectively balance strong reasoning and visual grounding with mobile-friendly efficiency, making real-world deployment of egocentric assistants far more practical.

To rigorously evaluate goal-oriented reasoning in this setting, we further introduce **Ego-IntentBench**, a new multimodal benchmark derived from Ego4D's Goal-Step and FHO annotations (Grauman et al., 2022). It comprises 471 clips and 15,853 procedural steps across two granularities, with a particular emphasis on complex cooking scenarios that require long-horizon task understanding. Compared to prior datasets like AssistQ (Wong et al., 2022) and MECCANO (Ragusa et al., 2023), Ego-IntentBench provides a more realistic testbed for intent modeling, fine-grained task decomposition, and multimodal reasoning in egocentric contexts.

Extensive experiments validate the effectiveness of Ego-VGA. On AssistQ, it achieves an 8.7% absolute gain in recall@1 over prior methods. On YouCook2, it outperforms the best baseline by 17.2 BLEU-1 and 7.6 METEOR points, while on MECCANO (RGB-only) it improves Mean Top-5 Recall by 20%. These results highlight Ego-VGA's strength in affordance reasoning, multimodal integration, and action–object understanding. On the more challenging Ego-IntentBench, despite overall performance drops, Ego-VGA surpasses LLM-based baselines like Qwen2.5-VL and MiniCPM-V4, demonstrating robust generalization for goal-directed, visually guided reasoning.

## 2 RELATED WORK

**AI Assistant.** Early AI assistants relied on NLP (Chowdhary, 2020) and rule-based dialogue, but lacked contextual reasoning. The emergence of LLMs such as GPT (Radford et al., 2018), LLaMA (Touvron et al., 2023a;b), and Gemini (Team et al., 2024) brought stronger comprehension, reasoning, and multi-turn dialogue, with multimodal variants (e.g., Flamingo (Alayrac et al., 2022), MiniGPT (Zhu et al., 2023)) enabling image-grounded interaction. Lightweight extensions such as MobileLLaMA (Wong et al., 2022) and LLaVA-mini (Zhang et al., 2025) further improved efficiency via compression, token reduction, and compact projectors, supporting real-time tasks on edge devices. More recently, embodied AI assistants like EgoGPT (Yang et al., 2025) and V-IRL (Yang et al., 2024) incorporated egocentric video, temporal modeling, and AR guidance, underscoring the demand for assistants that are multimodal, context-aware, and resource-efficient.

**Video Understanding.** Video understanding has evolved from 3D CNNs (C3D (Tran et al., 2015), I3D (Carreira & Zisserman, 2017)) and temporal aggregation networks (TSN (Wang et al., 2016), TRN (Zhou et al., 2018a)) to dual-pathway models like SlowFast (Feichtenhofer et al., 2019). Transformers (TimeSformer (Bertasius et al., 2021), ViViT (Arnab et al., 2021)) extended this to long-range dependencies but remained largely action-recognition centric. Joint video–language modeling shifted the paradigm: VideoGPT (Yan et al., 2021) and VideoBERT (Sun et al., 2019) tokenized video for text alignment, while MERLOT (Zellers et al., 2021) and MERLOT Reserve (Zellers et al., 2022) improved temporal reasoning through large-scale caption grounding. Recent MLLMs such as Video-LLaMA (Zhang et al., 2023b), InternVideo (Wang et al., 2022), and MiniCPM-v (Yao et al., 2024) integrate video tokens into LLMs, enabling advanced reasoning, video QA, and multi-turn dialogue, pushing generalization in complex video tasks.

**Datasets.** Benchmarks for procedural understanding span task execution, step reasoning, and causal modeling. Early datasets like YouCook2 (Zhou et al., 2018b) and OPRA emphasized cooking

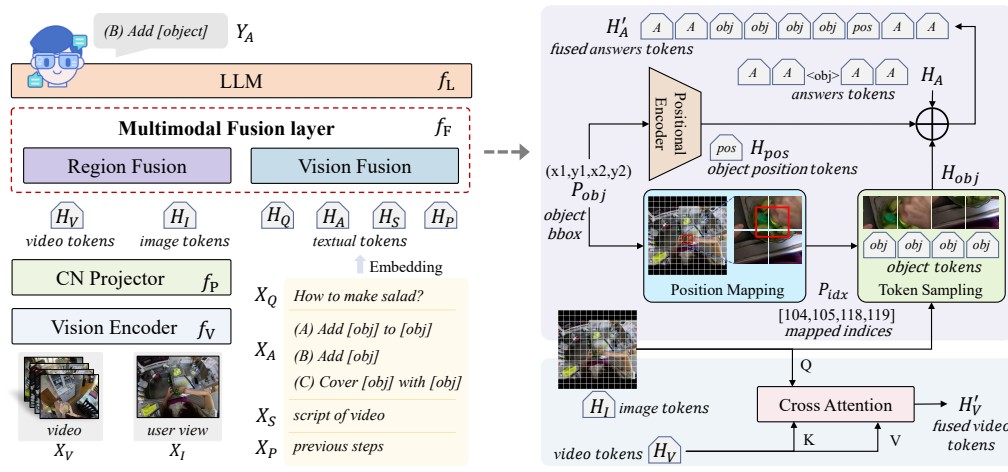

(a) model overall architecture      (b) process of region fusion (top) and vision fusion (bottom)

Figure 1: Illustrations of Ego-VGA. (a) model architecture; (b) proposed fusion layer workflow.

and affordance reasoning, while large-scale corpora such as HowTo100M (Miech et al., 2019) and CrossTask (Zhukov et al., 2019) supported weakly supervised pretraining and cross-task transfer. Most, however, remain observation-centric. Wearable devices spurred egocentric datasets: Epic-Kitchens (Damen et al., 2018; 2022), MECCANO, and Ego4D (Grauman et al., 2022) provided fine-grained actions, forecasting, and social interaction. Specialized benchmarks have since emerged: EgoSchema (Mangalam et al., 2023) for long-form QA, HiREST (Zala et al., 2023) for hierarchical reasoning, GazeVQA (Ilaslan et al., 2023) and EgoTextVQA (Zhou et al., 2025) for multimodal QA with gaze/text cues, EgoExoLearn (Huang et al., 2024a) and EgoMe (Qiu et al., 2025) for cross-view and personalization, and InstructionBench (Wei et al., 2025) for task decomposition and interactive instruction. Together, these datasets bridge procedural reasoning toward real-world AI assistants.

## 3 EGO-VGA

To begin with, we define the task of goal-oriented egocentric assistant. The task requires an egocentric AI assistant to ground procedural knowledge acquired from instructional videos into the user's current first-person visual context, and to generate the next action in which the manipulated objects are explicitly localized within the user's view. Formally, given an instructional video $X_V$, a situational image $X_I$ representing the user's current observation, a goal-oriented query $X_Q$, answers $X_A$, a script $X_S$, and previous steps $X_P$, the assistant must select the contextually appropriate next step $Y_A$ from a set of candidate answers. This task is particularly valuable as it enables seamless transfer of multi-step procedural knowledge from video demonstrations to real-world execution, thereby alleviating the cognitive burden of frequent attention switching. Prior AI assistants, however, were either text-only and lacked visual grounding, or relied on large multimodal models with prohibitive computational costs, limiting their practicality in real-world, resource-constrained scenarios.

### 3.1 OVERALL ARCHITECTURE

As shown in Fig. 1, Ego-VGA integrates four lightweight yet complementary modules: 1) a vision encoder for semantic feature extraction; 2) a compact normalized projector for modality alignment; 3) a multimodal fusion layer for deep vision–language interaction; and 4) a mobile-friendly LLM for contextual reasoning and task inference.

**Vision Encoder.** We adopt CLIP ViT-L/14 (Radford et al., 2021) as a unified backbone to encode user images $X_I$ and video frames $X_V$ into embeddings $F_I$ and $F_V$. Using a single pre-trained encoder ensures consistent cross-modal representations while reducing parameters and memory, forming an efficient foundation for downstream fusion.

**CompactNorm Projector.** To bridge modalities, we design a lightweight projector. It maps $F_I$ and $F_V$ into aligned textual embeddings $H_I$ and $H_V$, retaining spatial cues while lowering computation. The normalization mitigates instability (e.g., gradient explosion), leading to more robust training and faster convergence. This design improves both efficiency and deployability (see Sec. 3.3).

**Multimodal Fusion Layer.** We design a two-stage fusion mechanism. First, *vision fusion* applies cross-attention (Vaswani et al., 2017) between $H_I$ and $H_V$ to enrich video embeddings $H_V'$. Second, *region fusion* grounds object information by injecting object embeddings $H_{obj}$ and positional cues $H_{pos}$ into placeholder tokens within answers, producing $H_A'$. This full-spectrum integration couples perception with reasoning, enabling fine-grained localization and action prediction—one of Ego-VGA's core contributions (see Sec. 3.2).

**Language Model Backbone.** At the reasoning core, we adopt MobileLLaMA (Chu et al., 2023), an efficient LLaMA variant that balances compactness and reasoning ability. Conditioned on fused tokens $(H_I, H_V')$, queries $H_Q$, and partial answers $H_A'$, it generates responses or predicts next-step actions. This lightweight yet capable backbone makes Ego-VGA practical for real-world tasks.

We next elaborate on CompactNorm Projector and Fusion Layer which are our main contributions.

## 3.2 Multimodal Fusion Layer

The multimodal fusion layer $f_F$ integrates visual and textual signals through two complementary mechanisms: *region fusion* for object grounding and *vision fusion* for context-aware video understanding. Together, these modules establish a tighter coupling between perception and reasoning, which is essential for accurate action prediction.

**Region Fusion.** A central challenge in vision–language reasoning lies in grounding textual references (e.g., *"pizza"*, *"oven"*) to their corresponding visual counterparts. To address this, our region fusion module replaces object placeholders in the answer tokens $H_A$ with multimodally fused object tokens, yielding grounded answer representations $H_A'$. This ensures that generated instructions explicitly reference the relevant visual entities rather than relying on plain text.

Unlike prior approaches such as AssistQ, which rely on masked-button encodings, our method fuses object-level visual tokens with positional cues. Specifically, given object coordinates $P_{obj}$, we compute their indices $P_{idx}$ within the image tokens $H_I$, sample the corresponding object tokens $H_{obj}$, and encode the spatial information as positional tokens $H_{pos}$. The final grounded answer representation is then formed by substituting placeholders $[PLACE]$ in $H_A$ with the fused object tokens:

$$P_{idx} = f_{\text{PM}}(P_{obj}), \quad H_{obj} = f_{\text{TS}}(P_{idx}, H_I), \quad H_{pos} = f_{\text{PE}}(P_{obj}), \tag{1}$$

$$H_A' = f_{\text{F}}(H_A, H_I) = H_A[[PLACE] \rightarrow \text{Concat}(H_{obj}, H_{pos})], \tag{2}$$

where $f_{\text{PM}}$, $f_{\text{TS}}$, and $f_{\text{PE}}$ denote position mapping, token sampling, and positional encoding, respectively. The index mapping from feature patches to downsampled tokens follows:

$$n = \left\lfloor \frac{\lfloor m/W_{in} \rfloor}{S} \right\rfloor \cdot W_{out} + \left\lfloor \frac{m \bmod W_{in}}{S} \right\rfloor, \tag{3}$$

with $m, n$ as indices, $W_{in}, W_{out}$ as input/output side lengths, and $S$ as the downsampling stride (details in Appendix B.2).

This grounding mechanism is applied not only to the current answers but also to prior step descriptions $H_P$, resulting in grounded representations $H_P' = f_{\text{F}}(H_P, H_I)$. Other textual inputs, such as questions $H_Q$ or script tokens $H_S$, are used directly without additional grounding.

**Vision Fusion.** While region fusion grounds objects at the token level, vision fusion enhances temporal reasoning by aligning user images with video streams. User images $H_I$ typically capture high-quality snapshots of the operating context, whereas video streams $H_V$ often contain redundant or noisy frames. Directly reasoning over the full video is inefficient and may obscure critical cues. To mitigate this, we apply cross-attention between user images and video features, using $H_I$ as queries and $H_V$ as key-value pairs. This directs the model's focus toward video segments that are semantically consistent with the user's current state. The resulting fused video representation is:

$$H_V' = f_{\text{F}}(H_I, H_V), \tag{4}$$

which is subsequently fed into the LLM for downstream multimodal reasoning.

In this way, region fusion ensures precise vision–language grounding at the object level, while vision fusion distills temporal cues from noisy video streams. Their combination equips Ego-VGA with both fine-grained grounding and robust temporal reasoning, significantly strengthening its ability to handle complex embodied tasks.

## 3.3 COMPACTNORM PROJECTOR

To align high-dimensional visual features with the language embedding space under strict efficiency constraints, we design a *CompactNorm Projector* $f_P$. Our design builds upon LDPv2 (Chu et al., 2024), a lightweight feature downsampling module based on depthwise separable convolution, and extends it with an additional normalization layer for improved stability.

The projector integrates four key components: (1) *point-wise convolutions* for feature alignment, (2) *average pooling* for token reduction, (3) a residual PEG (Chu et al., 2021) module to enhance positional awareness, and (4) an additional *normalization layer* to regulate feature statistics and stabilize optimization. Its overall network architecture is illustrated in Fig. 4 in Appendix.

Given image features $F_I \in \mathbb{R}^{N \times D}$ and video features $F_V \in \mathbb{R}^{T \times N \times D}$, where $N = HW/P^2$ is the number of visual patches, $D$ the feature dimension, and $T$ the number of frames, the projector maps these inputs into the language embedding space:

$$H_I = f_P(F_I), \quad H_V = f_P(F_V) = T f_P(F_I), \tag{5}$$

where $H_I \in \mathbb{R}^{N \times D_{\text{model}}}$ and $H_V \in \mathbb{R}^{T \times N \times D_{\text{model}}}$, with $D_{\text{model}}$ denoting the dimension aligned to the language model.

Compared with traditional projectors such as Q-Former (Li et al., 2023) or MLP-based mappings (Chen et al., 2020; Caron et al., 2020), our CompactNorm design preserves spatial information while substantially reducing computation and memory usage, making it well-suited for lightweight multimodal frameworks. Importantly, we observe that the residual accumulation in the PEG module may cause unstable training or gradient explosion, particularly when the input sequence is long (many patches or frames) or the initial feature variance is high. To mitigate this, our normalization layer regulates feature magnitudes before fusion, effectively suppressing exploding gradients and accelerating convergence. As shown in Fig. 5 in Appendix, this modification yields smoother gradient dynamics and faster convergence compared with the baseline LDPv2. Thus, CompactNorm not only inherits the efficiency of lightweight projectors but also introduces robustness and stability crucial for real-world deployment on mobile or edge devices.

## 3.4 TRAINING PIPELINE

Our training pipeline builds upon MobileLLaMA (Chu et al., 2023), following the response-generation paradigm of LLaVA (Liu et al., 2023), but extends it to support *simultaneous* integration of video, image, and contextual textual inputs. This unified multimodal design allows Ego-VGA to reason not only from a single modality, but from complementary streams of user views, instructional videos, and textual context. Moreover, to exploit the inherently multi-step nature of egocentric tasks, we adopt a teacher-forcing strategy during training.

Given a video $X_V$, a user-view image $X_I$, a question $X_Q$, ground-truth answers $X_A$, previous steps $X_P$, and script $X_S$, the textual input at step $j$ is defined as:

$$X_T^j = \begin{cases} \text{Concat}(X_Q, X_A^j, X_S), & j = 1, \\ \text{Concat}(X_Q, X_A^j, X_S, X_P^{[1:j-1]}), & j > 1, \end{cases} \tag{6}$$

where $j$ indexes the task step. The input is encoded with fused visual features as:

$$H_T^j = f_F(X_T^j, H_I^j). \tag{7}$$

The training objective maximizes the likelihood of the answer sequence:

$$P\left(X_A^j \mid X_V, X_I^j, X_T^j\right) = \prod_{i=1}^{L} P_\theta\left(X_A^{j,[i]} \mid H_V', H_I^j, H_T^{j,[1:i-1]}\right). \tag{8}$$

**Stage 1: Pre-training.** The first stage focuses on *vision–language alignment*. As shown in Fig. 2a, we freeze both the vision encoder and the language model, training only the projection layer on AssistQ (Wong et al., 2022). Two tasks are used: (i) device recognition from complete videos, and (ii) caption generation for video segments. This stage encourages the projector to map visual embeddings into the language space while preserving semantics. The textual input $X_T$ is treated in a general form, with tokens $H_T = f_F(X_T)$.

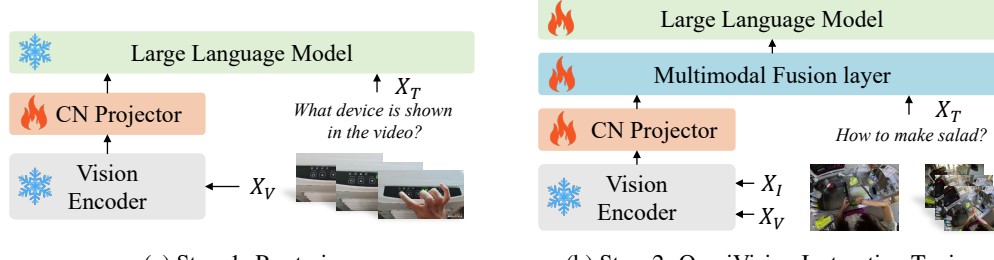

(a) Stage1: Pre-train  (b) Stage2: OmniVision Instruction Tuning

Figure 2: Illustrations of the Ego-VGA training strategy

**Stage 2: OmniVision Instruction Tuning.** The second stage equips the model with reasoning and instruction-following capability. Here, we freeze the vision encoder and fine-tune the language model, projector (initialized from Stage 1), and fusion layer (randomly initialized). Using the Affordance-centric Question-driven Task Completion (AQTC) task from AssistQ, the model learns to capture subtle task requirements, localize relevant objects, and generalize to unseen instructions.

This two-stage pipeline offers both stability and efficiency: Stage 1 ensures robust cross-modal alignment, while Stage 2 injects task-level reasoning and generalization. By jointly leveraging multi-step supervision and multimodal fusion, Ego-VGA achieves strong performance in context-rich egocentric settings while remaining lightweight and deployable.

## 4 EGO-INTENTBENCH

AssistQ dataset was originally designed for the Affordance-centric Question-driven Task Completion (AQTC) task, focusing on instructional videos of devices and their affordances. While useful, its scope is narrow and does not reflect the broader needs of contemporary egocentric AI assistants, particularly in goal-oriented domains like cooking, where tasks involve diverse objects, longer procedures, and more complex reasoning.

### 4.1 DATASET COLLECTION

To address this gap, we construct Ego-IntentBench, a new benchmark built on top of the Ego4D Goal-Step annotations and FHO bounding box labels. Ego-IntentBench extends AssistQ in three critical dimensions. See examples in Fig. 3.

- Domain expansion – from device-centric affordances to cooking tasks involving diverse utensils and ingredients;
- Increased complexity – longer videos with more fine-grained procedural steps, stressing long-term memory and multi-step reasoning;
- Richer supervision – integration of video narration and multimodal QA, enabling grounded reasoning and action guidance.

The dataset is organized by the goal–step hierarchy from Ego4D, supporting both coarse-grained (goal-level) and fine-grained (step-level) reasoning. Details are outlined below.

**User Situation.** Unlike AssistQ, where a single device image suffices to capture all manipulable elements, cooking tasks involve heterogeneous objects that vary by step. To reflect realistic assistant scenarios, we extract for each step a representative frame where all relevant objects appear and annotate it with bounding boxes from FHO. These step-level user view images provide more precise situational grounding.

**Annotation.** QA pairs are generated from Ego4D's category, description, and summary fields. Compared with device tasks, cooking steps involve fewer but more context-sensitive objects, requiring precise reasoning. To increase difficulty, distractor answers are drawn from other steps within the same task. We also incorporate the narration_text field as aligned video scripts, providing additional temporal and semantic cues.

**Quality Control.** To ensure reliability, all annotations undergo three rounds of human verification by independent annotators. This process validates image quality (e.g., motion blur, codec artifacts), bounding box correctness, and QA alignment with video content.

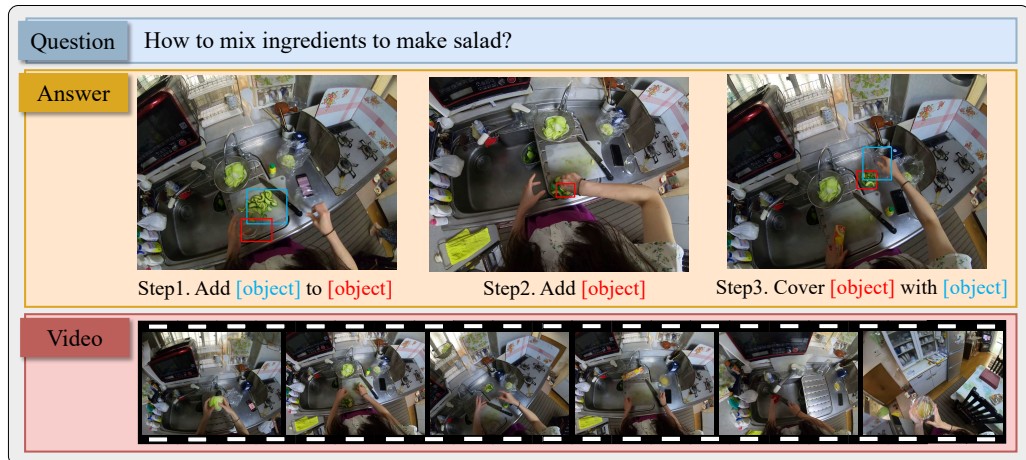

Figure 3: An example of Ego-IntentBench dataset

Table 1: Comparison with prior datasets. 'VG' denotes vision-guided data; Our dataset is vision-guided and multi-step, with longer videos and more steps, posing challenges for task understanding and planning.

| Dataset | Scenarios | View | Task | VG | Ans | Clips | Samples | Dur(s) | Steps |
|---|---|---|---|---|---|---|---|---|---|
| MovieQA (Tapaswi et al., 2016) | Movie | Exo | Factoid | × | Single | 408 | 14944 | 202.7 | |
| TVQA (Lei et al., 2018) | TV show | Exo | Factoid | × | Single | 21793 | 152545 | 76.2 | |
| OPRA (Fang et al., 2018) | Appliance | Both | Affordance | × | - | 11505 | 20774 | 8 | |
| AssistQ (Wong et al., 2022) | Appliance | Ego | Affordance | ✓ | Multi | 100 | 531 | 115 | 3.02 |
| Youcook2 (Zhou et al., 2018b) | Cooking | Exo | Procedural | × | Single | 2000 | 15400 | 316.2 | 7.72 |
| Epic-Kitchens (Damen et al., 2018; 2022) | Cooking | Ego | Procedural | × | - | 700 | 8997 | 514.29 | |
| MECCANO (Ragusa et al., 2023) | Industrial | Ego | Procedural | × | - | 20 | 8857 | 1247.4 | |
| Ego4D Goal-Step (Grauman et al., 2022) | Multiple | Ego | Goal-Oriented | × | Single | 851 | 47721 | 1560 | 23.82 |
| EgoTextVQA (Zhou et al., 2025) | Multiple | Ego | Goal-Oriented | × | Single | 1507 | 7064 | 101.7 | |
| InstructionBench (Wei et al., 2025) | Multiple | Both | Goal-Oriented | × | Single | 932 | 5000 | 282.95 | 10.39 |
| Ego-IntentBench (ours) | Cooking | Ego | Goal-Oriented | ✓ | Multi | 122 | 471 | 496.01 | **33.66** |

## 4.2 DATASET STATISTICS

Ego-IntentBench comprises 112 egocentric videos with an average duration of 496 seconds, encompassing 471 QA samples and 15,853 annotated steps, averaging around 33 steps per task. Compared with prior benchmarks (Table 1), Ego-IntentBench substantially increases scale and complexity, featuring videos nearly four times longer and ten times more steps than AssistQ, which significantly elevates the difficulty of long-horizon reasoning. Beyond device affordances, the dataset covers diverse goal-oriented tasks, reflecting more realistic and varied user scenarios. Each QA pair is enriched with multimodal supervision, including user-view images, bounding boxes, enabling models to perform vision-grounded, multi-step reasoning. Following the Goal-Step hierarchy, we split the data with an 8:2 train–validation ratio. Together, these characteristics make Ego-IntentBench a challenging, multimodal, and goal-driven benchmark for advancing egocentric AI assistants capable of long-term procedural understanding and context-aware guidance.

## 5 EXPERIMENT

We evaluated our model on the AssistQ, OPRA, YouCook2, MECCANO, and Ego-IntentBench datasets across instruction-following, vision-guided reasoning, object affordance understanding, fine-grained spatio-temporal reasoning, and complex scene comprehension. We also compared its execution efficiency with similarly sized multimodal models on AssistQ, demonstrating superior training and inference throughput under resource constraints. Experiments were run on a single NVIDIA RTX 4090 GPU.

### 5.1 MAIN RESULTS

**Results on AssistQ benchmark.** Table 2 reports results on the AssistQ AQTC benchmark. Traditional QA models (Q2A, Question2Function) achieve low Recall@1 (R@1), highlighting the challenge of grounding affordance-centric queries. Vision-language models (Q2F-VideoCLIP, Q2F-EgoVLP) substantially improve performance, reaching 78.4% and 78.7% R@1, showing the bene-

Table 2: Results of different methods on the AQTC task of the AssistQ dataset. 'Res.' denotes the resolution of preprocessed images and video frames.

| Method | Frames | Res. | Recall@1 | Recall@3 |
|---|---|---|---|---|
| Q2A (Wong et al., 2022) | 1 fps | 384 | 67.5 | 89.2 |
| Question2Function (Wu et al., 2022) | 1 fps | 384 | 62.6 | 87.5 |
| Q2F-VideoCLIP (Zhang et al., 2023a) | 1 fps | 224 | 78.4 | 93.8 |
| Q2F-EgoVLP (Chen et al., 2023b) | 1 fps | 336 | 78.7 | 93.4 |
| Qwen2.5-VL (Bai et al., 2025) | 8 frames | 336 | 84.4 | 86.1 |
| MiniCPM-V4 (Team et al., 2025) | 8 frames | 448 | 81.8 | 90.1 |
| Ego-VGA (ours) | 8 frames | 336 | **87.1** | **94.4** |

Table 3: Top-1 action recognition accuracy of different methods on the OPRA dataset.

| Method | Variants | Acc |
|---|---|---|
| Demo2Vec (Fang et al., 2018) | | 40.79 |
| Aformer (Chen et al., 2023a) | ViTDet-B-Aformer | 52.27 |
| | ResNet-50-Deconv | 48.93 |
| | ResNet-50-Aformer | 52.50 |
| InternVL2.5 (Chen et al., 2024) | | 49.13 |
| Qwen2.5-VL (Bai et al., 2025) | | 46.92 |
| Ego-VGA (ours) | | **52.68** |

Table 4: Mean Top-5 Recall for action anticipation on the MECCANO dataset. RGB and Depth capture appearance and geometry, Gaze encodes attention, and OBJ/Hands provide structured entity annotations.

| Model | RGB | Depth | OBJ | Gaze | Hands | MT5R |
|---|---|---|---|---|---|---|
| RULSTM (Furnari & Farinella, 2020) | ✓ | - | - | - | - | 22.88 |
| | ✓ | ✓ | - | - | - | 14.12 |
| | ✓ | - | ✓ | - | - | 29.03 |
| | ✓ | ✓ | ✓ | ✓ | - | 28.05 |
| | - | - | ✓ | ✓ | ✓ | 32.25 |
| | ✓ | ✓ | ✓ | ✓ | ✓ | 24.05 |
| Qwen2.5-VL | ✓ | - | - | - | - | **42.01** |
| InternVL2.5 | ✓ | - | - | - | - | **31.05** |
| Ego-VGA (ours) | ✓ | - | - | - | - | **42.78** |

fit of pretrained multimodal representations. Larger foundation models (Qwen2.5-VL, MiniCPM-V4) further boost accuracy via stronger language reasoning. Our method achieves the best results with 87.1% @1 and 94.4% Recall@3 (R@3), demonstrating superior query understanding, object grounding, and multi-step instruction generation, and establishing a new state-of-the-art for affordance-centric, vision-guided task execution.

**Results on OPRA benchmark.** Table 3 shows that early recurrent models struggle with fine-grained spatiotemporal dependencies, while adding temporal attention or motion cues improves accuracy. Aformer variants outperform recurrent baselines, and our model achieves the best results, highlighting its strength in affordance reasoning and structured temporal modeling for task-oriented video understanding.

**Results on YouCook2 benchmark.** On YouCook2, our model achieves 46.1 BLEU-1(B1), 14.1 BLEU-4(B4), and 19.4 METEOR(MET), consistently outperforming both the baseline GEPSAN and LLM-based approaches such as Qwen2.5-VL (Table 5). These results demonstrate the model's superior cross-view generalization and its ability to capture long-term dependencies, fine-grained interactions, and cross-modal alignment.

**Results on MECCANO benchmark.** Unlike prior works using Depth or Gaze, we focus on RGB-only inputs to match resource-constrained AI assistant scenarios. In this setting, our model achieves 42.78% mean top-5 recall (MT5R) (Table 4), nearly doubling the previous RGB-only state-of-the-art. This demonstrates strong generalization to both kitchen-style and industrial-like environments, where multi-step procedures and fine-grained hand–object interactions are essential.

**Results on Ego-IntentBench benchmark.** We evaluate LLM-based methods on the proposed Ego-IntentBench dataset (Table 6). Compared to their performance on AssistQ, all models show a performance drop: Qwen2.5-VL 77.7, MiniCPM-V4 78.2, and our Ego-VGA 78.5 in Recall@1. Notably, Ego-VGA surpasses both Qwen2.5-VL (+0.8) and MiniCPM-V4 (+0.3), demonstrating its robustness and strong generalization capability in complex goal-directed visual grounding tasks.

**Comparison of model efficiency.** We further evaluate the efficiency of Ego-VGA against Qwen2.5-VL and MiniCPM-V4 on the AssistQ dataset (Table 7). Despite having a comparable model size, Ego-VGA requires the lowest training FLOPs ($1.06 \times 10^{13}$), substantially lower than both baselines. In addition, it achieves the fastest per-step execution, with a training step time of 6.02s and an inference step time of 0.56s. These results indicate that Ego-VGA strikes a superior

Table 5: Results on the YouCook2 future anticipation task under the GePSAn protocol.

| Method | B1 | B4 | MET |
|---|---|---|---|
| Baseline (Sener et al., 2022) | 25.8 | 4.0 | 9.8 |
| GEPSAN (Abdelsalam et al., 2023) | 28.9 | 5.8 | 11.8 |
| InternVL2.5 (Chen et al., 2024) | 26.7 | 6.3 | 11.4 |
| Qwen2.5-VL (Bai et al., 2025) | 28.3 | 7.1 | 12.4 |
| Ego-VGA (ours) | **46.1** | **14.1** | **19.4** |

Table 6: Comparison of different methods on the Ego-IntentBench dataset.

| Method | Frames | Res. | Recall@1 |
|---|---|---|---|
| Qwen2.5-VL | 8 | 336 | 77.7 |
| MiniCPM-V4 | 8 | 448 | 78.2 |
| InternVL2.5 | 8 | 336 | 77.8 |
| Gemma3 | 8 | 336 | 76.9 |
| Ego-VGA (ours) | 8 | 336 | **78.5** |

Table 7: Comparison of model size and efficiency across different models.

| Model | Model Size | FLOPs | Train Time / Step(s) | Inference Time / Step(s) |
|---|---|---|---|---|
| Qwen2.5-VL (Bai et al., 2025) | 3B | $7.52 \times 10^{13}$ | 14.71 | 1.02 |
| MiniCPM-V4 (Team et al., 2025) | 4B | $4.83 \times 10^{13}$ | 35.71 | 2.75 |
| Ego-VGA (ours) | 3B | $\mathbf{1.06 \times 10^{13}}$ | **6.02** | **0.56** |

Table 8: Ablation study of projector. "PW" denotes point-wise, "AvgPool" denotes average pooling, and "Norm" denotes normalization.

| Projectors Architecture | R@1 | R@3 |
|---|---|---|
| PW + AvgPool + PEG | 77.5 | 89.4 |
| PW + AvgPool + Norm + PEG | 79.8 | 89.7 |
| PW + AvgPool + PEG + Norm | **82.2** | **91.7** |

Table 9: Ablation study of the proposed multimodal fusion layer.

| Region Fusion | Vision Fusion | R@1 | R@3 |
|---|---|---|---|
| × | × | 78.5 | 89.4 |
| ✓ | × | 78.8 | 88.4 |
| × | ✓ | 79.9 | 91.0 |
| ✓ | ✓ | **82.2** | **91.7** |

balance between computational cost and efficiency, making it particularly suitable for video understanding under resource-constrained conditions.

## 5.2 ABLATION STUDY

**Impact of Different Projector.** We conduct an ablation study to investigate the role of normalization in the projector design. As shown in Table 8, adding normalization substantially improves performance over the baseline LDPv2 architecture. Moreover, as illustrated in Fig. 5 of the appendix, normalization leads to smoother gradient dynamics and faster convergence. These findings demonstrate that normalization is a critical component for stable and efficient training, enabling our projector to retain the efficiency of lightweight designs while gaining robustness and stability essential for deployment on mobile or edge devices.

**Impact of Different Fusion Layer.** We further ablate the proposed multimodal fusion layer by disentangling region fusion and vision fusion. As shown in Table 9, disabling both fusion modules leads to the lowest performance, confirming their necessity. Region fusion alone yields marginal improvements in R@1 but slightly decreases R@3, suggesting that local alignment is insufficient for robust retrieval. In contrast, vision fusion alone provides a stronger gain, especially in R@3, highlighting the importance of visual integration. Combining both modules achieves the best overall results (82.2 R@1 and 91.7 R@3), indicating that region fusion and vision fusion are complementary and jointly crucial for effective multimodal alignment.

## 6 CONCLUSION

We present Ego-VGA, a lightweight multimodal egocentric assistant that provides goal-oriented visual guidance with high efficiency. By introducing a multimodal fusion layer for fine-grained vision–language grounding and temporal reasoning, alongside a lightweight downsampling projection and a compact LLM, Ego-VGA achieves strong performance while remaining deployable on mobile and edge devices. To support research in egocentric goal-oriented reasoning, we construct Ego-IntentBench, a challenging benchmark with fine-grained procedural annotations in complex tasks. Extensive experiments demonstrate that Ego-VGA outperforms state-of-the-art methods across multiple datasets, generalizes to third-person instructional and industrial scenarios, and excels under the increased difficulty of Ego-IntentBench. Our work establishes practical foundations for context-aware, multimodal egocentric AI assistants.

**Limitations.** The model is optimized for mobile deployment, balancing efficiency and multimodal reasoning, but may still face challenges under extreme resource constraints or strict low-latency requirements. Our dataset currently focuses on cooking; extending it to broader daily activities would better reflect real-world egocentric AI assistant scenarios and enhance its utility.

## ETHICS STATEMENT

This research was conducted in accordance with established ethical standards. All experiments use publicly available datasets released for academic purposes, which are anonymized and contain no personally identifiable information. In particular, our Ego-IntentBench dataset is constructed by integrating and reorganizing subsets of the open-source Ego4D dataset, strictly following its licensing and data usage policies. No new human or animal data were collected.

We acknowledge potential societal impacts of large multimodal models, such as biased predictions, unfair treatment of certain groups, or misuse of generated content. To mitigate these risks, our study is limited to scientific exploration and evaluation in controlled settings, and the models and code are released solely for research purposes under responsible AI guidelines.

## REPRODUCIBILITY STATEMENT

We have taken multiple measures to ensure the reproducibility of our results. The model architectures and training strategies are described in Section 3, while experimental setups, ablation studies, and evaluation metrics are detailed in Section 5, with additional implementation details provided in the Appendix. All datasets used in this study are publicly accessible, and the collection process and configuration of our proposed dataset are described in Section 4. Furthermore, we plan to release the code, trained model checkpoints, and the dataset as anonymous supplementary materials, enabling other researchers to reproduce our experiments under the same settings.

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

## A    THE USE OF LARGE LANGUAGE MODELS

During dataset construction, we employ GPT-4 (Ouyang et al., 2022) to generate questions conditioned on the goal/step categories and description fields from the Ego4D Goal-Step annotations (Grauman et al., 2022). Additionally, GPT-4 is used to extract target objects from the description fields, and their corresponding bounding box locations are retrieved from the FHO annotations. The complete procedure is detailed in the Appendix C.

## B    COMPACTNORM PROJECTOR

### B.1    PROJECTOR STRUCTURE

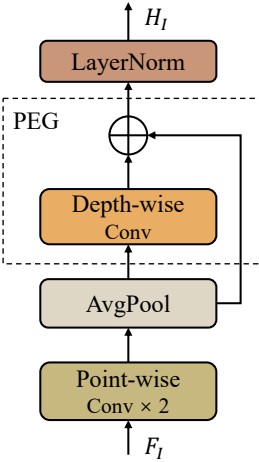

Figure 4: Architecture of the CompactNorm projector

Inspired by LDPv2 (Chu et al., 2023), we introduce an improved projection layer that enhances feature representation by combining channel transformation, spatial downsampling, positional encoding, and normalization.

The overall process begins with the input feature map ($F_I$) obtained from the backbone. Two successive point-wise convolutions are first applied to adjust channel dimensionality, either expanding or compressing the feature space while preserving spatial resolution, thereby facilitating effective channel integration. The downsampled representation is then produced via average pooling, which reduces spatial redundancy and provides compact contextual information. This step is particularly important for the subsequent Positional Encoding Generator (PEG), as it lowers computational overhead while highlighting spatial locality. At the core of the projection layer lies the PEG module, designed to inject positional inductive bias. It consists of a depth-wise convolution and a residual connection. The depth-wise convolution captures local spatial patterns across each channel independently, serving as dynamic positional encodings, while the residual connection directly adds the pooled features back to the convolutional output, ensuring residual learning and stable gradient propagation. The PEG-enhanced features are then fused with the output of the point-wise convolutions through residual addition, effectively combining high-resolution semantic information with localized positional cues. Finally, LayerNorm is applied to the fused representation, which stabilizes training and ensures consistent feature statistics across channels and spatial locations.

Overall, the design reflects four key principles: decoupling channel and spatial processing via point-wise and depth-wise convolutions, dynamic positional encoding through the PEG module, residual learning for efficient information flow, and normalization for training stability. Together, these components yield robust and discriminative feature representations tailored for downstream tasks.

## B.2 Mapped Index Calculation

Based on the aforementioned projection layer, a user view is mapped to a sequence of tokens in the textual embedding space. Our objective is to determine the token indices corresponding to an object, given its bounding box coordinates in the original user view. These indices are then used to extract the token subsequence representing the object in the textual space. The computational procedure for obtaining these indices is formally detailed as follows.

Let:

- $i$ denote the 1D index of the original input token, ranging from $0$ to $575$.
- $W_{\text{in}}$ be the width (or height) of the input feature map, i.e., $24$.
- $W_{\text{out}}$ be the width (or height) of the downsampled feature map, i.e., $12$.
- $S$ represent the downsampling stride, i.e., $2$.
- $j$ denote the 1D index of the downsampled token.

Step 1: Convert the 1D index to 2D coordinates

$$h_{\text{in}} = \left\lfloor \frac{i}{W_{\text{in}}} \right\rfloor \tag{9}$$

$$w_{\text{in}} = i \bmod W_{\text{in}} \tag{10}$$

Step 2: Compute the new 2D coordinates after downsampling

$$h_{\text{out}} = \left\lfloor \frac{h_{\text{in}}}{S} \right\rfloor = \left\lfloor \frac{\lfloor i/W_{\text{in}} \rfloor}{S} \right\rfloor \tag{11}$$

$$w_{\text{out}} = \left\lfloor \frac{w_{\text{in}}}{S} \right\rfloor = \left\lfloor \frac{i \bmod W_{\text{in}}}{S} \right\rfloor \tag{12}$$

Step 3: Map the new 2D coordinates back to a 1D index

$$j = h_{\text{out}} \cdot W_{\text{out}} + w_{\text{out}} \tag{13}$$

Step 4: Combine all steps to obtain the final mapping formula

$$j = \left\lfloor \frac{\lfloor i/W_{\text{in}} \rfloor}{S} \right\rfloor \cdot W_{\text{out}} + \left\lfloor \frac{i \bmod W_{\text{in}}}{S} \right\rfloor \tag{14}$$

## B.3 Projector Structure Ablation Experiment

Figure 5 compares two configurations of the projection layer: PW + AvgPool + PEG (orange) and PW + AvgPool + PEG + Norm (Ours) (blue). The additional normalization (Norm) in our design plays a crucial role in stabilizing training and improving performance.

**Gradient Norm.** The left plot shows the evolution of the $L_2$ norm of gradients. Without normalization, the gradient norm exhibits large fluctuations, particularly in the first 800 steps, indicating unstable optimization and susceptibility to gradient explosion. With normalization, the gradient norm remains consistently lower and converges smoothly toward zero, highlighting improved stability and faster convergence.

**Recall@1.** The right plot presents the Recall@1 trajectory. The baseline improves slowly and converges to 0.78. In contrast, our design with normalization achieves faster gains from the beginning and stabilizes around 0.84, yielding a notable improvement of approximately 6 percentage points.

**Conclusion.** Incorporating normalization into the projection layer is essential for stabilizing gradients, accelerating convergence, and ultimately learning more discriminative and generalizable representations, as evidenced by the substantial performance gap in Recall@1.

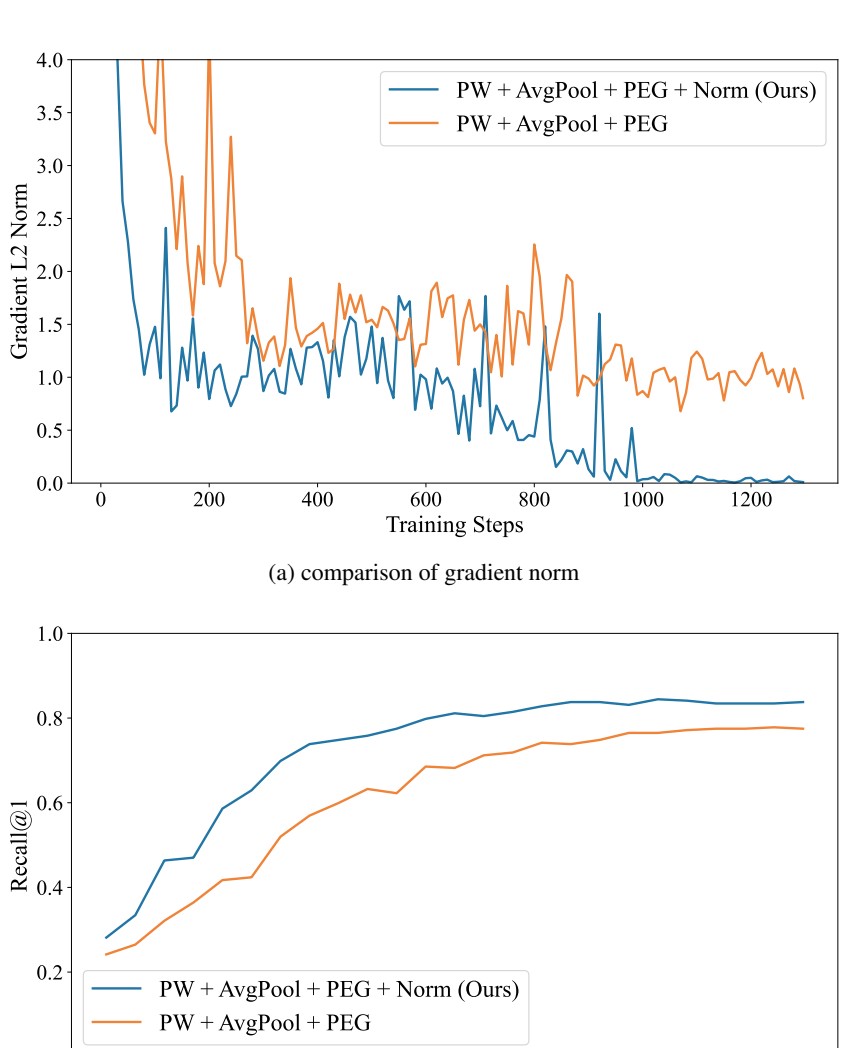

(a) comparison of gradient norm

(b) comparison of Recall@1

Figure 5: Comparison of the additional normalization layer. "PW" denotes point-wise, "AvgPool" denotes average pooling, and "Norm" denotes normalization.

## C  DATASET COLLECTION

To generate task-oriented questions for our dataset, we employ GPT-4 conditioned on the goal and step categories as well as the description fields from the Ego4D Goal-Step annotations. Specifically, GPT-4 is prompted to produce natural and concise questions that resemble those a user would ask an AI assistant. Each question explicitly asks how to perform the task described by the corresponding category and description. By providing the step-by-step summary of the correct answer, the generated questions become more specific, accurate, and actionable, closely reflecting realistic scenarios. The prompt used for this process is illustrated in Figure 6.

---

You are a question generator.
Given the following task information, generate one natural question that a user would ask an AI assistant.

Category: {category}
Description: {description}
Summary (list of steps): {summary}

Requirements:
1. The generated question should sound like a user asking an AI assistant for help.
2. The correct answer must be exactly the list of steps in the summary.
3. The question must directly ask how to perform the task described in the category and description.
4. The question should be short, clear, and natural.
5. Only return the question text. Do not include explanations, additional text, or JSON formatting.

---

Figure 6: The prompt used with GPT-4 to generate goal-oriented questions.

To identify target objects in our dataset, we further employ GPT-4 to extract object nouns from the description fields of Ego4D Goal-Step annotations. The extraction process retains only head nouns and focuses exclusively on objects directly involved in the described actions. Subsequently, a double-check is performed using the object taxonomy list from the FHO annotations to ensure consistency and validity. Finally, the spatial locations of these objects are obtained from the FHO annotations in the form of bounding boxes. The prompt used for this extraction procedure is illustrated in Figure 7.

---

You are an information extraction assistant.
Task: Extract the nouns of the objects involved in the action from the following sentence.
- Return only the head nouns (not adjectives).
- If there are multiple objects, return them separated by commas.
- Do not include explanations.

Examples:
Input: Add lemon paste
Output: paste

Input: Cut onion and tomato
Output: onion, tomato

Input: Mix water, flour and sugar
Output: water, flour, sugar

Now extract for the following:
{step description}

---

Figure 7: The prompt used with GPT-4 to extract manipulated objects.

## D  SAMPLES OF THE EXPERIMENTAL DATASETS

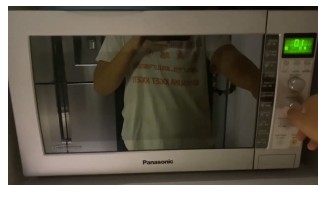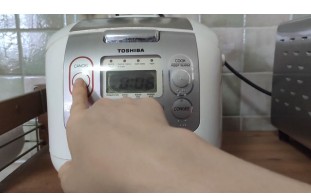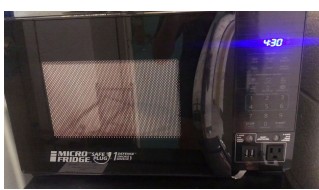

**(a)** Challenging examples from the AssistQ dataset

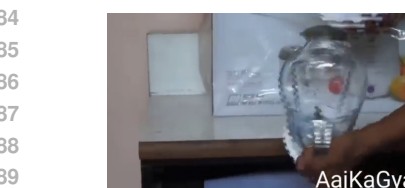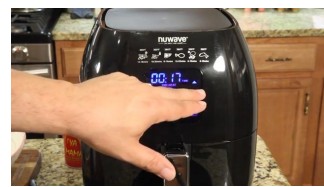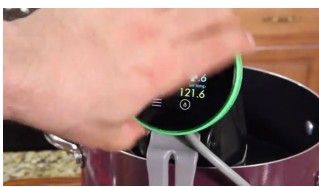

**(b)** Challenging examples from the OPRA dataset

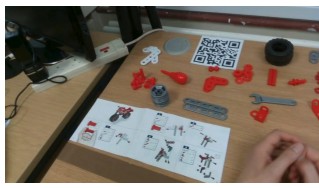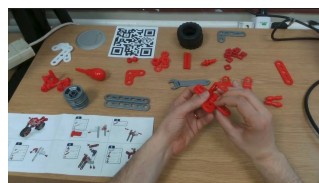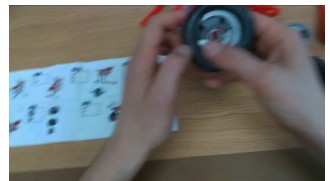

**(c)** Challenging examples from the MECCANO dataset

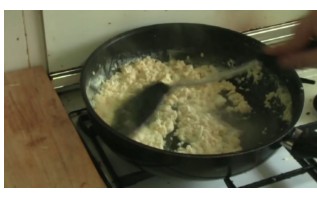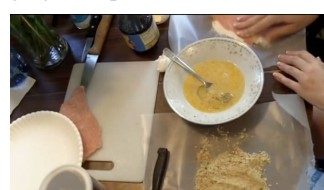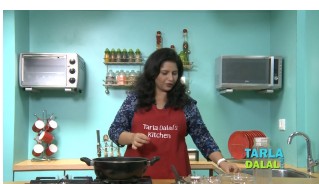

**(d)** Challenging examples from the YouCook2 dataset

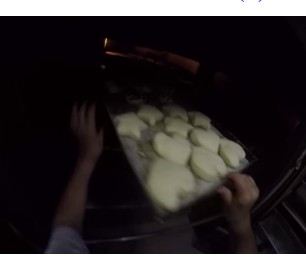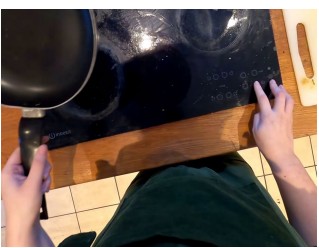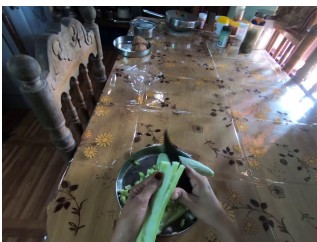

**(e)** Challenging examples from the Ego-IntentBench dataset

**Figure 8:** Challenging examples from the experimental datasets

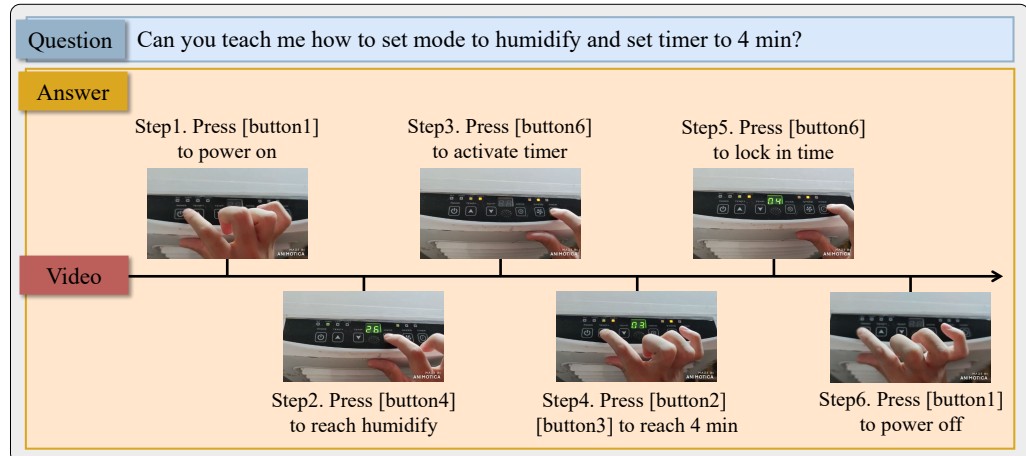

**Figure 9:** An example of AssistQ dataset

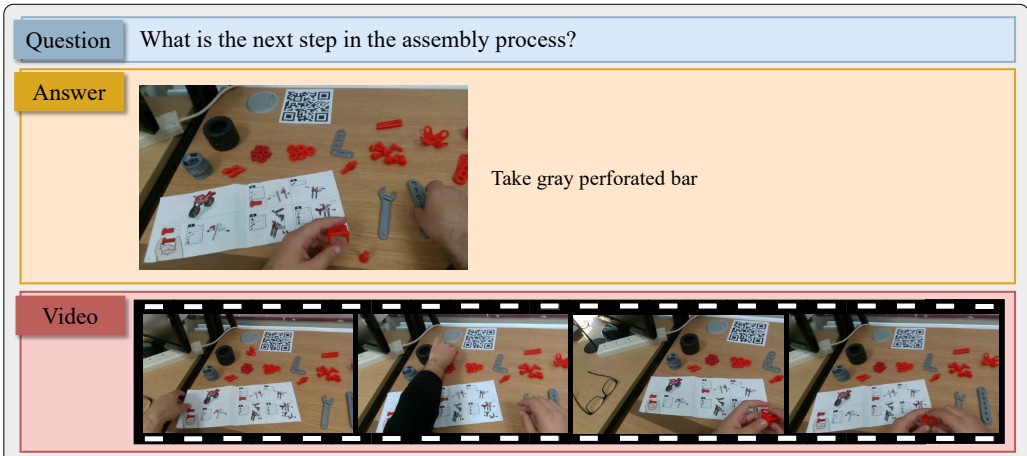

**Figure 10:** An example of MECCANO dataset

Fig. 9 illustrates a representative example from AssistQ, where the model must teach a user how to set the humidify mode and configure a 4-minute timer. This example clearly demonstrates that AssistQ tasks involve long procedures with strict action ordering, as the answer requires identifying and sequencing six distinct operations—powering on the device, adjusting the mode, activating the timer, selecting the target duration, locking the setting, and powering off. Each action depends on the successful completion of the previous one, forming a tightly constrained temporal structure.

Moreover, the example highlights AssistQ's long-horizon, multi-step activities that rely on causal dependencies across extended time spans. For instance, the timer cannot be locked (Step 5) unless it has been activated (Step 3) and the duration has been set (Step 4). The model must track visual changes across widely separated frames, maintain memory of non-visible intermediate states (e.g., whether the timer mode is already active), and infer causal relationships between user actions and the resulting interface updates. Together, these properties show that AssistQ requires deep temporal reasoning far beyond simple frame-level pattern matching.

In MECCANO, assembling small mechanical components inherently leads to frequent hand-object occlusions, requiring the model to maintain a persistent memory of object states even when key parts temporarily disappear from view. As illustrated in Fig. 10 , the target component—the gray perforated bar—is fully or partially occluded by the user's hands across multiple frames, forcing the model to infer its location and state over time rather than relying on instantaneous visual evidence.

# E    MULI-STEP EXPERIMENT

**Table 10:** Model performance on AssistQ and Ego-IntentBench under a multi-step QA setting

| Model | Metric | AssistQ | | Ego-IntentBench | |
|---|---|---|---|---|---|
| | | single-step | multi-step | single-step | multi-step |
| Ower2.5-VL | R@1 | 84.4 | 82.1 ↓ 2.3 | 77.7 | 68.4 ↓ 9.3 |
| | R@3 | 86.1 | 84.1 ↓ 2.0 | 87.3 | 77.0 ↓ 10.3 |
| MiniCPM-V4 | R@1 | 81.8 | 80.8 ↓ 1.0 | 78.2 | 70.5 ↓ 7.7 |
| | R@3 | 90.1 | 85.4 ↓ 4.7 | 83.8 | 75.1 ↓ 8.7 |
| Ego-VGA (ours) | R@1 | 87.1 | 85.4 ↓ 1.7 | 78.5 | 67.4 ↓ 11.1 |
| | R@3 | 94.4 | 93.7 ↓ 0.7 | 97.0 | 90.9 ↓ 6.1 |

We extended our evaluation to a multi-step QA setting by incorporating the previous step's Question and Answer as part of the input for the current step. As shown in Table 10, all models exhibit a performance decline when transitioning from single-step to multi-step evaluation, which is expected given the increased complexity of reasoning across sequential steps. Notably, Ego-VGA demonstrates the strongest robustness: its R@3 decreases by only 0.7 points on AssistQ and by 6.1 points on the more challenging Ego-IntentBench benchmark, indicating a superior ability to leverage contextual information from prior steps. In contrast, baseline models such as Owen2.5-VL and MiniCPM-V4 show substantially larger drops—particularly on Ego-IntentBench—highlighting that effective multi-step reasoning remains difficult for existing methods.

A closer examination reveals that multi-step QA introduces additional uncertainty arising from error accumulation, evolving visual states, and ambiguous user intentions. These factors primarily affect the model's confidence sharpness—namely, whether the correct option is ranked first—rather than disrupting the overall ranking structure. As a result, models are more likely to mis-rank the top-1 candidate, leading to a noticeable reduction in R@1. Nevertheless, Ego-VGA maintains strong semantic alignment between user queries and video content. Even when multi-step noise slightly perturbs the top prediction, the correct answer typically remains among the top candidates, resulting in only marginal degradation in R@3.

Overall, these findings underscore the importance of incorporating historical QA context and demonstrate the effectiveness of our approach in handling the additional reasoning complexity inherent to multi-step interactive scenarios.

# F    CROSS-ATTENTION EXPERIMENTS IN THE VISION FUSION

**Table 11:** Performance comparison on the AssistQ dataset under different numbers of cross-attention layers.

| Attention Layers | Recall@1 | Recall@3 |
|---|---|---|
| 1 | 82.8 | 91.9 |
| 2 | 79.0 | 91.3 |
| 3 | 75.5 | 88.5 |
| 4 | 79.3 | 90.9 |

The results in Table 11 indicate that adding more layers does not improve performance; instead, it often leads to degradation. We hypothesize that deeper cross-attention introduces unnecessary temporal mixing and weakens the alignment between the current egocentric view and the most relevant video frames. Additional layers may also amplify noise and hinder the model's ability to maintain a stable egocentric alignment signal. These findings suggest that, under our task setting, a shallow Vision Fusion module is not only computationally optimal but also empirically the most effective

**Table 12:** Performance comparison on the AssistQ dataset under different numbers of cross-attention heads.

| Attention Heads | Recall@1 | Recall@3 |
|---|---|---|
| 1 | 82.8 | 91.9 |
| 8 | 82.1 | 92.1 |
| 16 | 82.3 | 91.4 |
| 40 | 81.9 | 90.4 |

The ablation results in Table 12 show that varying the number of cross-attention heads has only a marginal impact on performance. Using a single head achieves competitive results, with Recall@1 of 82.8 and Recall@3 of 91.9. Increasing the number of heads to 8 or 16 does not lead to consistent improvements, and further enlarging the head count to 40 even results in slight performance degradation. This trend suggests that, given the high dimensionality of both image and video-frame features (2560), additional heads may introduce unnecessary parameter fragmentation and optimization instability without providing meaningful benefits. Overall, the results indicate that a lightweight single-head cross-attention design is sufficiently expressive for aligning user-view queries with video-frame context in this task.

