# OpenReview forum: "Ego-VGA: A Compact Multimodal Assistant for Egocentric Video–Grounded Reasoning"
_ICLR.cc/2026/Conference — Submitted to ICLR 2026_

### Official Review · Reviewer_pJYS · 2025-10-26

**Soundness:** 3
**Presentation:** 2
**Contribution:** 2
**Rating:** 6
**Confidence:** 4

**Summary:**

This paper tackles a key trade-off in egocentric AI: large models are too resource-intensive for on-device deployment, while lightweight models lack robust visual grounding. The authors propose Ego-VGA, a lightweight multimodal assistant designed for efficient, goal-oriented visual guidance.
Core innovations include a novel Multimodal Fusion Layer—using Region Fusion for fine-grained grounding and Vision Fusion for temporal reasoning—and a lightweight CompactNorm Projector for stable, efficient multimodal alignment.
The authors also contribute Ego-IntentBench, a new, challenging benchmark derived from Ego4D for complex procedural tasks.
Experimental results show Ego-VGA achieves state-of-the-art performance on multiple benchmarks while being significantly more computationally efficient than comparable models like Qwen2.5-VL and MiniCPM-V4.

**Strengths:**

1. The paper exhibits an outstanding emphasis on computational efficiency. In particular, the proposed Ego-VGA model achieves competitive or superior accuracy compared to existing 3B–4B vision-language models, while drastically reducing computational cost (about one-seventh of Qwen2.5-VL in training FLOPs).

2. The introduction of Ego-IntentBench is a clear contribution. The authors address a genuine gap in current benchmarks by emphasizing long-horizon, complex egocentric tasks with detailed annotations. I believe this dataset will be a valuable asset for future research in egocentric intent understanding.

**Weaknesses:**

1. A limitation of the new benchmark (Ego-IntentBench) is its narrow scope. By focusing only on cooking, it lacks diversity and cannot fully test how well a model generalizes to other goal-oriented tasks. Adding other daily activities, like repairing or cleaning, would make it a more comprehensive and valuable benchmark.

2. The paper claims that the model is suitable for complex scenarios requiring long-horizon task understanding (e.g., Ego-IntentBench). However, its training paradigm (Section 3.4) appears to adopt a teacher-forcing strategy. As shown in Equation (6), for steps where j>1, the model takes the ground-truth previous steps XP[1:j−1] as input. This training method hides a critical issue in real-world settings — error accumulation. In deployment, the model does not have access to the true previous actions; it must rely on its own (potentially incorrect) predictions as inputs for subsequent steps. The paper provides no experiments demonstrating the robustness of Ego-VGA under such a more realistic auto-regressive inference setup.

**Questions:**

1.Where is the Pobj in the Multimodal Fusion Layer coming from? If Pobj is sourced from an external object detector, then its significant computational cost appears to be omitted from the efficiency analysis in Table 7, potentially undermining the argument for mobile deployment.

2.The comparison on Ego-IntentBench (Table 6) seems potentially unfair, as generative models like Qwen2.5-VL are evaluated on a discriminative multiple-choice task. How were these generative baselines adapted for this setting? Were they fine-tuned on the Ego-IntentBench training set like Ego-VGA? Clarifying this is crucial to determine whether Ego-VGA’s advantage reflects stronger reasoning or just task-specific adaptation.

3.The description of the Region Fusion mechanism in Section 3.2 seems inconsistent. The text mentions that the method “replaces object placeholders in the answer tokens HA,” while Formula (2) defines the process as Concat(HA,Hobj,Hpos). Since replacement and concatenation represent fundamentally different operations, could you clarify the exact integration mechanism of object (Hobj) and position (Hpos​) tokens into the answer sequence (HA)? Specifically, are these tokens concatenated to the sequence end, or inserted/replaced at the corresponding placeholder positions?

4.Could you provide more architectural details on the cross-attention module used in the "Vision Fusion" stage? For instance, how many attention layers and heads does it contain, and is it followed by feed-forward networks?

---

> ### Author Response · Authors · 2025-11-21
>
> Thank you for the insightful and positive comments! In the following, we provide our point-by-point response and hope our response helps address your concerns. We also look forward to the subsequent discussion which may further help solve the current issues.
>
> ### Q1. Limitation of Ego-IntentBench dataset scenarios
>
>
> Thanks. We would like to address your concern from seveal aspects.
>
> First, cooking, as one of the most ubiquitous everyday activities, involves diverse object interactions, rich action patterns, and complex multi-step structures. Its clear procedural organization, hierarchical goal structure, and strong temporal dependencies closely align with the core challenges of goal-oriented egocentric tasks such as intent reasoning, goal prediction, and step-based question answering, making it a highly representative and canonical form of goal-oriented egocentric activity.
>
> Second, we would like to clarify that our dataset design is not arbitrary, but inherently constrained by the landscape of existing large-scale, high-quality multi-step egocentric datasets. The publicly available corpora that provide sufficiently rich annotations and long-horizon multi-step interactions—such as EPIC-Kitchens, HD-EPIC, CaptainCook4D, and Ego4D Goal-Step—are predominantly cooking-oriented. This also partially indicates that other works also agree the importance of cooking, aligning with our settings. As a result, constructing a multi-step dataset that is both realistic and scalable inevitably requires leveraging these sources, which in turn limits the feasible scenario domain to the cooking-related category.
>
> Third, although our dataset is limited to the cooking domain, it offers several significant advantages. As shown in Table 1, although our dataset contains a relatively small number of clips, each clip is significantly longer on average (top-5 among compared datasets). This indicates that the overall dataset size remains substantial. More importantly, each clip contains far more steps than existing multi-step video datasets. Our design prioritizes quality and structural complexity over raw quantity: each clip captures rich intent transitions, step dependencies, and fine-grained human–object interactions, all of which are essential for evaluating goal-oriented reasoning. Compared with mainstream multi-step benchmarks, Ego-IntentBench offers substantially higher temporal density and task complexity, making it a high-quality and challenging benchmark. Furthermore, our dataset is complementary to other long-video but low-step datasets, on which we also evaluate our model to obtain a broader and more comprehensive assessment.
>
> Finally, we agree with the reviewer that focusing on a single domain does not fully represent all goal-oriented egocentric tasks. However, for other everyday scenarios such as repair and cleaning, there is currently a lack of high-quality, open-source multi-step datasets. Collecting such data would require crawling relevant videos from the web, generating operation videos in virtual environments, or conducting real-world recordings. All of these approaches are highly time-consuming and cannot be reasonably completed within the limited rebuttal period. Therefore, we consider this as an important direction for future work.

---

> ### Author Response · Authors · 2025-11-21
>
> ### Q2. On teacher-forcing training and error accumulation in multi-step reasoning
>
> Thank you for the suggestion. We adopt this setting in our paper to ensure fair comparison: our work follows the [LOVEU Challenge](https://sites.google.com/view/loveucvpr23/track3) protocol, where the task is formulated as a single-step QA problem, i.e., using teacher-forcing during inference. This setting is also widely used in the community, like our compared baselines. Auto-regressive inference, by contrast, corresponds to a true multi-step QA scenario, which requires reliable step-level grounding, temporal alignment, and stable short-horizon reasoning—all of which critically depend on the correctness and robustness of the single-step QA component. Therefore, single-step QA problem, as the fundamental building block of multi-step QA, possesses substantial intrinsic research value.
>
> Following your suggestion, we further include experiments that more closely reflect real-world deployment conditions. Specifically, we re-evaluate Ego-VGA under an auto-regressive inference setting on both the AssistQ and Ego-IntentBench datasets to assess the model’s robustness. The newly added experimental results are presented in the table below and have been included in Appendix E of the paper.
>
> **Table 10. Model performance on AssistQ and Ego-IntentBench under a multi-step QA setting**
> | Model              | Metric | AssistQ (single-step) | AssistQ (multi-step) | Ego-IntentBench (single-step) | Ego-IntentBench (multi-step) |
> | ------------------ | ------ | --------------------- | -------------------- | ----------------------------- | ---------------------------- |
> | **Ower2.5-VL**     | R@1    | 84.4                  | 82.1 ↓ 2.3           | 77.7                          | 68.4 ↓ 9.3                   |
> |                    | R@3    | 86.1                  | 84.1 ↓ 2.0           | 87.3                          | 77.0 ↓ 10.3                  |
> | **MiniCPM-V4**     | R@1    | 81.8                  | 80.8 ↓ 1.0           | 78.2                          | 70.5 ↓ 7.7                   |
> |                    | R@3    | 90.1                  | 85.4 ↓ 4.7           | 83.8                          | 75.1 ↓ 8.7                   |
> | **Ego-VGA (ours)** | R@1    | 87.1                  | 85.4 ↓ 1.7           | 78.5                          | 67.4 ↓ 11.1                  |
> |                    | R@3    | 94.4                  | 93.7 ↓ 0.7           | 97.0                          | 90.9 ↓ 6.1                   |
>
>
> While all models experience some performance drop compared to single-step evaluation, Ego-VGA exhibits the smallest decrease, particularly in R@3. This indicates that even when multi-step noise slightly perturbs the top prediction, the correct answer typically remains among the top candidates. These results validate that our model effectively leverages historical QA context and demonstrates superior robustness and contextual integration ability in realistic auto-regressive inference tasks.

---

> ### Author Response · Authors · 2025-11-21
>
> ### Q3. The source of $P_{obj}$ in the Multimodal Fusion Layer
>
> To ensure a fair comparison, we follow the experimental setting adopted in the previous LOVEU Challenge and subsequent high-performance works, such as Q2F-VideoCLIP and Q2F-EgoVLP. Specifically, the $P_{obj}$ used in our Multimodal Fusion Layer is directly provided by the dataset as ground-truth button bounding-box annotations. In other words, $P_{obj}$ is not produced by an external object detector during inference, and therefore incurs no additional computational cost in Table 7.
>
> ### Q4. Clarification on the fairness of baseline results in table 6
>
> In Table 6, we adopt the dataset split defined in Section 4.2 for Ego-IntentBench. All baseline models are fine-tuned on the same training set following their officially recommended procedures. To ensure a fair comparison, we further optimized key hyperparameters—including learning rate and weight decay—toward better performance for each model. Evaluation is conducted on the identical validation split. This protocol guarantees that the comparison fairly reflects each model’s reasoning ability under their best-supported adaptation strategies, while respecting the training paradigms for which they were designed.
>
> ### Q5. Discrepancy between the textual description and the mathematical formulation
>
> We sincerely thank the reviewer for the careful reading. Indeed, there is a typo in the original paper: the expression ${Concat}(H_A, H_{obj}, H_{pos})$ in Formula (2) is inaccurate.
>
> As illustrated in Figure 1, we embed object placeholders $[\mathit{PLACE}]$ into the answer tokens $H_A$ to specify the locations at which object information should be inserted. Consistent with the description in Section 3.2, the object tokens $H_{obj}$ and positional tokens $H_{pos}$ are concatenated to form the content that replaces these placeholders, i.e., the complete object feature token sequence. The corrected formula is as follows:
>
> $\begin{equation}
> 	H_A' = f_\text{F}(H_A, H_I) = H_A[[\mathit{PLACE}] \rightarrow \operatorname{Concat}(H_{obj}, H_{pos})],
> \end{equation}$
>
> We have updated the formula in the paper to resolve the inconsistency between the textual description and the original formula.
>
> ### Q6. Cross-attention design in vision fusion
>
> Our vision fusion module adopts a lightweight single-layer cross-attention design tailored for egocentric video reasoning. The query (Q) corresponds to the user-view image features, while the key and value (K/V) are extracted from all video frames. All inputs are projected to a shared hidden dimension of 2560, and we employ a single-head scaled dot-product attention mechanism without stacking additional layers.
>
> This minimalist design avoids over-parameterization and focuses attention capacity on modeling the alignment between the egocentric view and the global video context, it balances model performance and computational efficiency, making it particularly suitable for deployment on resource-constrained devices.
>
> To further validate the effectiveness of this configuration, we conducted additional experiments on AssistQ dataset with varying numbers of cross-attention layers and attention heads. The results, presented in Tables 11 and 12 in Appendix F.
>
> **Tables 11. Comparison of different attention layers**
> | Attention Layers | Recall@1 | Recall@3 |
> | ---------------- | -------- | -------- |
> | 1                | 82.8     | 91.9     |
> | 2                | 79.0     | 91.3     |
> | 3                | 75.5     | 88.5     |
> | 4                | 79.3     | 90.9     |
>
> **Tables 12. Comparison of different attention heads**
> | Attention Heads | Recall@1 | Recall@3 |
> | --------------- | -------- | -------- |
> | 1               | 82.8     | 91.9     |
> | 8               | 82.1     | 92.1     |
> | 16              | 82.3     | 91.4     |
> | 40              | 81.9     | 90.4     |
>
> As shown in the results above, increasing the number of layers or heads either reduces R@1 or offers no consistent gain in R@3. Single-layer attention avoids noise accumulation across multi-step reasoning, while a single head preserves the full semantic capacity of high-dimensional features for robust cross-view alignment. Together, these results validate that our minimalist design is both effective and stable.

---

> ### Comment · Reviewer_pJYS · 2025-11-26
>
> The rebuttal has addressed my questions. I will keep my positive rating.

---

> > ### Author Response · Authors · 2025-11-26
> >
> > Many thanks. We are glad our response can help you better understand our work, and address your concerns. We will spend more effort and time to solve the issues mentioned. Thank you for your insightful and positive comments again!!

---

### Official Review · Reviewer_TgWA · 2025-11-01

**Soundness:** 3
**Presentation:** 2
**Contribution:** 3
**Rating:** 4
**Confidence:** 3

**Summary:**

This paper introduces Ego-VGA, a lightweight and efficient multimodal assistant designed for egocentric video-grounded reasoning on mobile devices. It addresses the critical trade-off between large, high-performance models that are too resource-intensive and compact models that lack strong visual grounding. The core of Ego-VGA is a novel multimodal fusion layer, which performs Region Fusion to ground textual instructions with specific objects in the user's view and Vision Fusion to distill relevant temporal cues from noisy video streams. To further enhance efficiency, the model uses a CompactNorm projector for stable modality alignment and a compact LLM (MobileLLaMA) for reasoning. The authors also introduce Ego-IntentBench, a new challenging benchmark for complex, goal-oriented cooking tasks. Experiments show that Ego-VGA achieves state-of-the-art results on multiple datasets, including AssistQ, YouCook2, and MECCANO , and outperforms larger models on the new Ego-IntentBench , all while demonstrating superior computational efficiency.

**Strengths:**

1. The paper tackles a well-defined and significant problem: the need for a practical, efficient, and high-performing egocentric assistant.
2. The two-pronged Multimodal Fusion Layer is a key strength. The Region Fusion module's approach of directly injecting grounded object and position tokens into answer placeholders  is an explicit and effective method for fine-grained vision-language grounding. The Vision Fusion module is an intelligent solution for handling noisy video streams by using the current user view as a cross-attention query to filter temporal context.
3. The introduction of Ego-IntentBench is a significant contribution.

**Weaknesses:**

1. As the authors acknowledge in their limitations , Ego-IntentBench is currently focused *exclusively* on cooking scenarios. While this is a complex domain, it is not representative of all goal-oriented egocentric tasks. The clip number of Ego-IntentBench is also small.
2. The "Vision Fusion" component, while effective, is a single cross-attention layer  that primarily filters video frames based on the current user view. This mechanism may be insufficient for tasks requiring deep temporal reasoning, memory of non-visible object states, or understanding complex causal relationships across long time gaps.

**Questions:**

Please refer to the weaknesses.

---

> ### Author Response · Authors · 2025-11-21
>
> Thank you for the insightful and valuable comments! In the following, we provide our point-by-point response and hope our response helps address your concerns. We also look forward to the subsequent discussion which may further help solve the current issues.
>
> ### Q1. Limitations of Ego-IntentBench dataset scenarios and clip number
>
> Thanks. We would like to address your concern from seveal aspects.
>
> First, cooking, as one of the most ubiquitous everyday activities, involves diverse object interactions, rich action patterns, and complex multi-step structures. Its clear procedural organization, hierarchical goal structure, and strong temporal dependencies closely align with the core challenges of goal-oriented egocentric tasks such as intent reasoning, goal prediction, and step-based question answering, making it a highly representative and canonical form of goal-oriented egocentric activity.
>
> Second, we would like to clarify that our dataset design is not arbitrary, but inherently constrained by the landscape of existing large-scale, high-quality multi-step egocentric datasets. The publicly available corpora that provide sufficiently rich annotations and long-horizon multi-step interactions—such as EPIC-Kitchens, HD-EPIC, CaptainCook4D, and Ego4D Goal-Step—are predominantly cooking-oriented. This also partially indicates that other works also agree the importance of cooking, aligning with our settings. As a result, constructing a multi-step dataset that is both realistic and scalable inevitably requires leveraging these sources, which in turn limits the feasible scenario domain to the cooking-related category.
>
> Third, although our dataset is limited to the cooking domain, it offers several significant advantages. As shown in Table 1, although our dataset contains a relatively small number of clips, each clip is significantly longer on average (top-5 among compared datasets). This indicates that the overall dataset size remains substantial. More importantly, each clip contains far more steps than existing multi-step video datasets. Our design prioritizes quality and structural complexity over raw quantity: each clip captures rich intent transitions, step dependencies, and fine-grained human–object interactions, all of which are essential for evaluating goal-oriented reasoning. Compared with mainstream multi-step benchmarks, Ego-IntentBench offers substantially higher temporal density and task complexity, making it a high-quality and challenging benchmark. Furthermore, our dataset is complementary to other long-video but low-step datasets, on which we also evaluate our model to obtain a broader and more comprehensive assessment.
>
> Finally, we agree with the reviewer that focusing on a single domain does not fully represent all goal-oriented egocentric tasks. However, for other everyday scenarios such as repair and cleaning, there is currently a lack of high-quality, open-source multi-step datasets. Collecting such data would require crawling relevant videos from the web, generating operation videos in virtual environments, or conducting real-world recordings. All of these approaches are highly time-consuming and cannot be reasonably completed within the limited rebuttal period. Therefore, we consider this as an important direction for future work.

---

> ### Author Response · Authors · 2025-11-21
>
> ### Q2. About the single cross-attention
>
> Many thanks. We would like to address your concern from seveal aspects.
>
> First, the goal of our work is to develop a lightweight multimodal assistant that provides goal-oriented visual guidance with high efficiency. Accordingly, we intentionally avoid designing a large model or incurring high inference costs. Our design choices are guided by this principle and are purposefully made to achieve the desired efficiency and effectiveness. Specifically, by using the user view as a compact query and the video frames as KV, the single-layer single-head cross-attention performs demand-driven temporal selection, allowing the model to attend only to the most relevant frames and actions. This design dramatically reduces the effective video token count, avoids unnecessary multi-layer propagation, and yields efficient yet strong temporal and causal reasoning.
>
> Moreover, regarding the task types highlighted by the reviewers, we emphasize that the datasets used in our evaluation already cover these challenging settings. For example, tasks in AssistQ and YouCook2 involve long procedures with strict action ordering, demanding deep temporal reasoning (Fig. 9 in Appendix D); MECCANO requires tracking object states when key components are occluded (Fig. 10 in Appendix D); AssistQ and Ego-IntentBench include long-horizon, multi-step activities that rely on recognizing causal dependencies across extended time spans (Fig. 9 in Appendix D and Fig. 3). Across all these scenarios, our model—with a single-layer cross-attention design—achieves strong performance while preserving a compact model size.
>
> In addition, we compare our approach with models that incorporate more sophisticated mechanisms. For instance, Qwen2.5-VL employs absolute temporal encoding and dynamic-resolution processing, enabling precise modeling of temporal relations and complex causal chains in long videos. MiniCPM-V4 leverages an efficient 3D-Resampler to aggressively compress video tokens and handle high frame rates, thereby capturing subtle motion variations and enhancing temporal reasoning. Despite these advanced techniques, our method still demonstrates competitive or superior performance on challenging benchmarks. For example, on AssistQ, our model achieves a higher score (87.1 vs. 81.8) while using a smaller model size and lower FLOPs (Table 7).
>
> Overall, despite the structural simplicity of our Vision Fusion module, our experiments show that combining single-layer cross-attention with the strong reasoning ability of MobileVLM V2 is sufficient to attain high performance on goal-oriented video QA. The model effectively aligns visual cues, captures essential temporal context, and performs multi-step reasoning—all within a lightweight architecture.
>
> To further address your concern regarding the fusion mechanism, we further include an ablation study evaluating different numbers of cross-attention layers and heads on the AssistQ dataset, as shown in Tables 11 and 12 in Appendix F.
>
> **Tables 11. Comparison of different attention layers**
> | Attention Layers | Recall@1 | Recall@3 |
> | ---------------- | -------- | -------- |
> | 1                | 82.8     | 91.9     |
> | 2                | 79.0     | 91.3     |
> | 3                | 75.5     | 88.5     |
> | 4                | 79.3     | 90.9     |
>
> **Tables 12. Comparison of different attention heads**
> | Attention Heads | Recall@1 | Recall@3 |
> | --------------- | -------- | -------- |
> | 1               | 82.8     | 91.9     |
> | 8               | 82.1     | 92.1     |
> | 16              | 82.3     | 91.4     |
> | 40              | 81.9     | 90.4     |
>
> As shown in the results above, increasing the number of layers or heads either reduces R@1 or offers no consistent gain in R@3. Single-layer attention avoids noise accumulation across multi-step reasoning, while a single head preserves the full semantic capacity of high-dimensional features for robust cross-view alignment. Together, these results validate that our minimalist design is both effective and stable.

---

> ### Author Response · Authors · 2025-11-27
>
> Thank you once again for your thoughtful review and valuable comments! With only seven days remaining before the end of the discussion phase, we would greatly appreciate your feedback on our revised responses. Please let us know whether our clarifications adequately address your concerns. If any issues remain unresolved or if further explanation is needed, we would be grateful for your additional guidance.
>
> Best regards, ﻿
>
> Authors

---

### Official Review · Reviewer_XKLE · 2025-11-10

**Soundness:** 3
**Presentation:** 3
**Contribution:** 3
**Rating:** 6
**Confidence:** 3

**Summary:**

This paper focuses on the problem that current egocentric VLMs are either resource consuming or weak in vision capabilities. It proposes Ego-VGA, a lightweight framework that is more efficient in training/inference compared to models with the same size. To further evaluate the model, the paper introduced Ego-IntentBench, a benchmark evaluates long-horizon understanding.

**Strengths:**

1. the design of Ego-VGA successfully achieves a much lower FLOPs compared to Qwen2.5-VL-3B and MiniCPM-V4-4B
2. Ego-VGA achieves the best performance in the reported results

**Weaknesses:**

1. The real-world application contains extreme environments such as low light, occlusion, and dynamic interference. It is better to evaluate the model's robustness in these environments.
2. Lacking of experiments on multi-term interaction.
3. There should be more baseline models to show the strength of proposed model.

**Questions:**

N/A

---

> ### Author Response · Authors · 2025-11-21
>
> Thank you for the insightful and positive comments! In the following, we provide our point-by-point response and hope our response helps address your concerns. We also look forward to the subsequent discussion which may further help solve the current issues.
>
> ### Q1. Evaluating model robustness under extreme scenarios
>
> Thank you for the suggestion. For goal-oriented egocentric AI assistants, robustness under extreme real-world conditions—such as low light, occlusion, and dynamic interference—is indeed crucial. Motivated by these factors, we carefully selected several datasets that naturally exhibit such challenges. Following your advice, we additionally provide visual examples in Appendix D to highlight the presence of low-light conditions, occlusions, dynamic interference, camera shake, motion blur, and other difficult scenarios.
>
>  - **AssistQ** consists of real-world, daily-life egocentric videos that naturally include numerous challenging conditions, such as button occlusion due to hand placement, narrow or partially hidden views, faint buttons under poor lighting, and motion blur caused by egomotion or out-of-focus cameras (Fig. 8(a)). These lead to frequent occurrences of low-light scenes, heavy occlusions, motion blur, hand interference, and rapid user actions.
>  - **OPRA** focuses on inferring object functionality from manipulation behaviors and similarly contains substantial challenges including object occlusion and motion blur resulting from fast actions (Fig. 8(b)).
>  - **MECCANO**, a multimodal egocentric dataset collected in industrial-like environments (Fig. 8(c)), exhibits severe illumination variations, viewpoint changes, partial occlusions (especially by hands), spatial constraints, and motion blur induced by rapid operations.
>  - **YouCook2** features highly diverse kitchen environments with complex lighting, steam and smoke interference, object occlusions, and rapid cooking motions (Fig. 8(d)).
>  - **Ego-IntentBench**, derived from the large-scale Ego4D Goal-Step dataset, primarily contains cooking-related indoor activities. The videos frequently involve low-light or uneven illumination, substantial occlusions, camera shake, dynamic disturbances, and motion blur (Fig. 8(e)).
>
> Collectively, these datasets cover a broad spectrum of extreme scenarios, which is why many prior works adopt them as evaluation benchmarks. In comparison, our evaluation protocol includes significantly more datasets than most existing baselines. For instance, GEPSAN reports results only on YouCook2, while StillFast evaluates solely on Ego4D.
>
> While the above datasets already incorporate diverse challenging conditions, we agree that evaluating under even more extreme real-world scenarios would further strengthen robustness analysis. We plan to include additional datasets featuring such extreme conditions in future work. However, collecting and curating these data requires substantial time, and due to the tight rebuttal timeline, we leave this as an important direction for future exploration.

---

> ### Author Response · Authors · 2025-11-21
>
> ### Q2. Multi-term interaction
>
> Based on our understanding, multi-term interaction mainly corresponds to multi-turn QA. In such a task, it requires designing a multimodal egocentric model equipped with capabilities such as multi-turn instruction following, context memory, planning, and reasoning, as highlighted in prior studies [1,2]. This is a highly challenging yet promising research direction. However, the mainstream task setup in existing benchmarks still adopts a single-step QA formulation, where the ground truth from the previous step is used as the input to the next step. For fairness, our work follows the [LOVEU Challenge](https://sites.google.com/view/loveucvpr23/track3) setting, where the task is defined as a single-step QA problem and is widely used for evaluation in many works like our baselines. Another reason we currently focus on single-step QA is that it serves as the fundamental building block of multi-step QA. Multi-step QA requires reliable step-level grounding, temporal alignment, and stable short-horizon reasoning, all of which depend on the correctness and robustness of the single-step QA component.
>
> To further address the reviewer’s concern, we extended our evaluation on the AssistQ and Ego-IntentBench datasets to a multi-step QA setting by feeding the previous step’s question and answer as part of the input for the next step, together with the current question. The newly added experimental results are shown in the table below and have also been included in Appendix E of the revised manuscript.
>
> **Table 10. Model performance on AssistQ and Ego-IntentBench under a multi-step QA setting**
> | Model              | Metric | AssistQ (single-step) | AssistQ (multi-step) | Ego-IntentBench (single-step) | Ego-IntentBench (multi-step) |
> | ------------------ | ------ | --------------------- | -------------------- | ----------------------------- | ---------------------------- |
> | **Ower2.5-VL**     | R@1    | 84.4                  | 82.1 ↓ 2.3           | 77.7                          | 68.4 ↓ 9.3                   |
> |                    | R@3    | 86.1                  | 84.1 ↓ 2.0           | 87.3                          | 77.0 ↓ 10.3                  |
> | **MiniCPM-V4**     | R@1    | 81.8                  | 80.8 ↓ 1.0           | 78.2                          | 70.5 ↓ 7.7                   |
> |                    | R@3    | 90.1                  | 85.4 ↓ 4.7           | 83.8                          | 75.1 ↓ 8.7                   |
> | **Ego-VGA (ours)** | R@1    | 87.1                  | 85.4 ↓ 1.7           | 78.5                          | 67.4 ↓ 11.1                  |
> |                    | R@3    | 94.4                  | 93.7 ↓ 0.7           | 97.0                          | 90.9 ↓ 6.1                   |
>
> Although all models show some degradation under the multi-step setting, Ego-VGA exhibits the smallest drop—especially on R@3. This suggests that, despite noise introduced across steps, the correct answer generally remains among the top candidates, confirming that our model effectively exploits historical QA context and maintains superior robustness in multi-step reasoning.
>
> ```
> [1] A Survey on Multi-Turn Interaction Capabilities of Large Language Models
> [2] Beyond single-turn: A survey on multi-turn interactions with large language models
> ```
>
>
> ### Q3. More baseline models
>
> Following your suggestion, we have incorporated additional baselines—Qwen2.5-VL (2025) and InternVL2.5 (2024)—in comparisons on the MECCANO dataset (Table 4), and added InternVL2.5 and Gemma3 (2025) on the Ego-IntentBench dataset (Table 6), with the results summarized below.
>
> **Table 4. Additional comparative results on the MECCANO dataset**
> | Model              | RGB | Depth | OBJ | Gaze | Hands |    MT5R   |
> | ------------------ | :-: | :---: | :-: | :--: | :---: | :-------: |
> | **Qwen2.5-VL**     |  ✓  |   -   |  -  |   -  |   -   | 42.01 |
> | **InternVL2.5**    |  ✓  |   -   |  -  |   -  |   -   | 31.05 |
> | **Ego-VGA (ours)** |  ✓  |   -   |  -  |   -  |   -   | **42.78** |
>
>
> **Table 6. Additional comparative results on the Ego-IntentBench dataset**
> | Method         | Frames | Res. | Recall@1 |
> | -------------- | :----: | :--: | :------: |
> | InternVL2.5    |    8   |  336 |   77.8   |
> | Gemma3         |    8   |  336 |   76.9   |
> | Ego-VGA (ours) |    8   |  336 | **78.5** |
>
> The results demonstrate that our method still outperforms these newly added baselines on their respective datasets, highlighting the effectiveness of our model in action understanding and complex intent reasoning.
>
> Here, we present the results that have been completed. Due to time constraints, experiments on the remaining datasets with additional baselines have not yet been conducted, but we plan to extend our evaluations to include more baselines in future work.

---

### Author Response · Authors · 2025-12-03
**Summary Comment**

Dear ACs,

We would first like to thank the ACs for their dedicated effort and thoughtful consideration.

During the rebuttal phase, we provided detailed, point-by-point responses to all reviewer comments. Reviewers XKLE and pJYS offered largely positive assessments, highlighting that our method achieves competitive or superior performance with significantly higher computational efficiency, and acknowledging the value of the proposed Ego-IntentBench dataset. We thoroughly addressed their concerns regarding model robustness under extreme scenarios and multi-term interaction capability, adding new multi-term interaction experiments, incorporating more baseline models, and clarifying technical details such as the source of $P_{obj}$ and the fairness of table 6 baselines. Notably, Reviewer pJYS explicitly stated that their concerns had been resolved and maintained a positive rating.

Reviewer TgWA acknowledged our effective multimodal fusion design and the value of introducing a new benchmark, but raised two concerns:
 - **Dataset limited to cooking scenarios and the number of clips being relatively small.** We clarified that the cooking domain intrinsically possesses strong task structure and high procedural complexity, making it particularly well aligned with the reasoning capabilities required for goal-oriented egocentric assistants. Importantly, cooking was not chosen arbitrarily: it is the most feasible and representative domain among existing high-quality multi-step egocentric video datasets—such as EPIC-Kitchens, HD-EPIC, and Ego4D Goal-Step—which also predominantly focus on cooking activities. Although our dataset contains fewer clips, each clip is substantially longer, includes a greater number of steps, and exhibits more complex task structures. As a result, our benchmark offers significantly higher temporal density and step complexity compared with existing multi-step datasets. We also agree on the importance of multi-scenario coverage; however, repair and cleaning currently lack high-quality, open-source, multi-step egocentric datasets. Collecting such data is time-consuming, and we therefore highlight multi-scenario expansion as a key direction for future work.

 - **Concern that using only a single-layer cross-attention for vision fusion may be insufficient for deep temporal reasoning, maintaining memory of non-visible object states, or modeling long-range causal dependencies.** We first emphasize that our design goal is to build a lightweight multimodal assistant that provides goal-oriented visual guidance with high efficiency. The types of challenging tasks the reviewer mentioned are already extensively covered in our experiments. To address this point more concretely, and we added more dataset examples in the revised version. Moreover, our experiments demonstrate that this minimalist fusion design achieves competitive or even superior performance compared to larger and more complex models such as Qwen2.5-VL and MiniCPM-V4 on these tasks. To further alleviate concerns regarding the fusion mechanism, we added additional ablation studies showing that our minimal cross-attention layer is both effective and stable across a diverse set of scenarios and reasoning types.

We fully respect the decision to revert the reviews for fairness. This note is provided solely to assist the ACs in interpreting the current reviews in the context of the revised manuscript.

Best Regards,

Authors

---

### Meta-Review · Area_Chair_RzeS · 2026-01-05

**Summary:**

This paper received mixed evaluations, with Reviewer XKLE and Reviewer pJYS assigning scores of 6  and Reviewer TgWA assigning a score of 4.

While reviewers acknowledge the model’s computational efficiency and the value of the Ego-IntentBench benchmark, several concerns were raised. Reviewer XKLE noted the limited evaluation of robustness under challenging real-world conditions, the lack of analysis on multi-turn interactions, and insufficient baseline comparisons. Reviewer TgWA questioned the generality of the benchmark, given its focus on cooking scenarios and limited number of clips, as well as the methodological design and the adequacy of the Vision Fusion module for long-term reasoning. Reviewer pJYS further raised concerns about the narrow scope of Ego-IntentBench, the mismatch between training and inference (teacher forcing versus autoregressive deployment), the fairness of baseline comparisons, unaccounted computational costs (e.g., object detection), and unclear architectural details.

**Reviewer Concerns:**

For Reviewer XKLE, the authors provided additional evaluations of model robustness under extreme scenarios, as well as expanded analyses of multi-turn interactions with additional baselines.

For Reviewer TgWA, the authors provided explanations on the limitations of the Ego-IntentBench dataset in terms of scenario diversity and clip count, along with clarification of the projector and model design.

For Reviewer pJYS, the authors also discussed the limitations of the Ego-IntentBench dataset, clarified the fairness of the baseline results reported in Table 6, and provided further details on the model architecture and design.

**Reviewer Scores:**

Although Reviewer pJYS indicated that the rebuttal addressed their questions and maintained a positive rating (6), I largely agree with the concerns raised by Reviewer TgWA (score 4) on two key points.

First, Ego-IntentBench has notable limitations in scenario diversity, as it focuses exclusively on cooking scenes and is therefore not representative of the full range of goal-oriented egocentric tasks; moreover, the number of clips （#122） and samples （#471） is limited. Second, the model places strong emphasis on projector design but lacks comparisons with several relevant recent methods, including LDPv2 (Arxiv), Honeybee (CVPR2024),  LLava-Mini(ICLR2025) and TokenPacker (IJCV2025). The relatively simple design appears insufficient for tasks that require deep temporal reasoning, long-term memory of non-visible object states, or modeling complex causal relationships over extended time horizons, which are crucial capabilities for egocentric scenarios.

---

### Decision · Program_Chairs · 2026-01-26

Reject